# First Study of *Pharingomyia picta* and *Cephanemyia auribarbis* in Wild Populations of Red Deer (*Cervus elaphus*) in Portugal

**DOI:** 10.3390/ani12151896

**Published:** 2022-07-26

**Authors:** Rui Miranda, João Serejo, Jesús M. Pérez, José Aranha, Carlos Venâncio, Madalena Vieira-Pinto

**Affiliations:** 1Department of Veterinary Science, University of Trás-os-Montes and Alto Douro (UTAD), Quinta de Prados, 5001-801 Vila Real, Portugal; 2Veterinary Office, Idanha-a-Nova Town Hall, 6060-163 Idanha-a-Nova, Portugal; jmqsp14@gmail.com; 3Department of Animal Biology, Plant Biology and Ecology, Campus Las Lagunillas s/n, University of Jaén, 23071 Jaén, Spain; jperez@ujaen.es; 4Centre for the Research and Technology of Agro-Environmental and Biological Sciences (CITAB), and Inov4Agro-Institute for Innovation, Capacity Building and Sustainability of Agri-Food Production, University of Trás-os-Montes and Alto Douro (UTAD), Quinta de Prados, 5000-801 Vila Real, Portugal; j_aranha@utad.pt (J.A.); cvenanci@utad.pt (C.V.); 5Department of Forestry Sciences and Landscape Architecture (CIFAP), University of Trás-os-Montes and Alto Douro (UTAD), 5001-801 Vila Real, Portugal; 6Department of Animal Science, University of Trás-os-Montes and Alto Douro (UTAD), 5001-801 Vila Real, Portugal; 7Veterinary and Animal Research Centre (CECAV), University of Trás-os-Montes and Alto Douro (UTAD), 5001-801 Vila Real, Portugal; 8Associate Laboratory for Animal and Veterinary Sciences (AL4AnimalS), University of Trás-os-Montes and Alto Douro (UTAD), 5001-801 Vila Real, Portugal

**Keywords:** *Pharyngomyia picta*, *Cephenemyia auribarbis*, oestrosis, nasopharyngeal myiases, red deer, *Cervus elaphus*

## Abstract

**Simple Summary:**

This study aimed to provide the first epidemiological data on nasopharyngeal myiasis in wild red deer from Portuguese territory. This parasitosis consists of the infestation of the host nasopharynx by Diptera larvae. In this study, the first scientific report in Portugal, a high level of prevalence (50.6%) and mean intensity of parasitism (11.38 larvae) were found. The following two larvae species were identified: *Pharyngomyia picta* and *Cephenemyia auribarbis*. The life cycle of this species can be summarised in the following series of stages: one adult stage, three larval stages, and one pupa stage. The larvae are the parasitic form, living and growing inside the host. With their spicules and hooks, they progress into the host’s nasal cavities causing sinusitis, sneezing, purulent mucous exudates, dyspnea, and swallowing. In addition, they can also cause neurologic symptomatology derived from erratic larvae migrations. Therefore, it would be important to develop further studies to estimate the sublethal effect of this parasitosis, for example, investigate whether it interferes with the roar and the reproductive performance, or if it leads to losses in carcass quality. For now, many doubts remain.

**Abstract:**

Nasopharyngeal myiasis in red deer has never been studied in Portugal. For this reason, from December 2015 to February 2016 a study was derived on hunted red deer in Idanha-a-Nova county to evaluate the Diptera larvae presence. During the initial examination on the spot, the larvae was analysed at the nasopharynx. When present, larvae were collected for further species identification. The infestation prevalence was 50.6 ± 7.61% (CI 95%), and the mean parasitism intensity was 11.38 larvae per affected host. Two larvae species were found, *Pharyngomyia picta* and *Cephenemyia auribarbis*, both on single or concomitant infection, with the first species having a higher prevalence and mean infestation intensity than the second. The larvae’s prevalence was significantly higher (X^2^ = 4.35; *p* = 0.0370) in males (62.16 ± 11.05%) than in females (41.30 ± 10.06%). Within age groups, younger animals showed a higher prevalence. This study shows the presence of *P. picta* and *C. auribarbis* larvae in the wild red deer population in Portugal for the first time. The high parasitic prevalence and mean intensity highlight the importance of this parasitosis in this wild species. For this reason, more scientific research is required to accurately assess its prevalence in other geographic areas, and evaluate the risk factors as well effects of this parasitosis on the deer population.

## 1. Introduction

Nasopharyngeal myiasis (Diptera: Oestridae: Oestrinae) results from the infestation of the host’s nasal cavity and throat with bot flies’ larvae, which affects Artiodactyls and Equidae [1]. Every member of this taxonomic group is an obligatory parasite, which means that larvae need living hosts and their respective tissue and secretions for their successful development [2,3]. All wild ruminant species hunted in Portugal (red deer, fallow deer, mouflon, and roe deer) are susceptible species to this parasitosis [4]. There are nine genders in this taxonomic group, but only two affect the cervids, *Cephenemyia* and *Pharyngomyia*, and they are distributed throughout the Holarctic and Palearctic regions, respectively [1,5]. The larvae of these flies are often found in *Cervus*, *Capreolus*, *Moose*, *Dama*, *Rangifer*, and *Odocoileus* [6,7]. Red deer (*Cervus elaphus*) are the focus of this study, and it is noteworthy that the larval species affecting these large game animals are *Cephenemyia auribarbis* and *Pharyngomyia picta* [8].

The life cycle of these species presents one adult stage, three larval stages, and one pupa phase [9]. The first larval stage is very dependent on environmental conditions, requiring exposure to air and contact with the body temperature of the animal to become active [10]. Afterwards, L1 larvae present a positive thermotropism and a negative phototropism that impels them to progress into the nasal/oral cavities of the host [11]. Using its spicules and hooks, they avoid expulsion by the movements and sneezing of the host [1,10], and move to the nasopharynx region where they develop into L2 and later into L3 [12,13,14]. When L3 larvae complete their development, they migrate to outside the host, where they look for a safe place on the ground, which will allow them to continue their ontogenesis until they become a pupa. Finally, after a few weeks, the cycle closes when adult individuals emerge.

The larvae development period in the host can be very variable, and one or two cycles can occur per year. Adults tend to remain active in the warmer months [5]. In addition, the development period of each larva may be different. This is motivated by the overcrowding of larvae in a limited space, particularly at the level of pharyngeal recesses, with larvae competing for the space they occupy, and also influenced by the host’s immune response [9]. De la Fuente et al. [13] performed a continuous study over a year in Toledo (Spain) and found that larval stages have a winter profile, from November to March, and the period of greatest intensity occurred from December to February.

Pathogenesis and symptomatology are better described in domestic species as expected since they cause damage to livestock production, as well because monitoring this parasitosis is easier than in wild species [1]. The larvae, as they progress into the nasal cavities with the aid of their spicules and hooks, and when they are fixed at the level of the nasopharynx through their oral apparatus, end up injuring the mucosa and promoting an inflammatory response in the host [12]. Some adverse effects associated with infestation by larvae are described in cervids, including eight loss, discomfort, rhinitis, sinusitis, sneezing, nasal discharge, dyspnea, and difficulty swallowing [1,3,11,15,16,17,18]. Moreover, it is also possible to cause the death of the host, and some authors suggest that there is a possible sublethal effect on deer [3,8]. In addition, neurologic symptomatology derived from erratic migrations of larvae can occur, such as walking in circles, motor discoordination, and disorientation [13]. These effects are dependent on the host susceptibility and the parasitism intensity.

There are several studies on the presence of nasopharyngeal myiasis in red deer in Europe (Poland; Spain; Hungary; Scotland; Austria) [15]. However, according to the authors’ knowledge, there is no published information on the distribution, presence and prevalence of nasopharyngeal myiasis in red deer in Portugal. For this reason, and due to the potential adverse effects associated with infestation by these larvae, the main objectives of this study were to evaluate the prevalence and intensity of infestation of dipteran larvae in wild red deer hunted in Portugal.

## 2. Materials and Methods

The study was performed in Idanha-a-Nova county, located in Castelo Branco district, in the east of the central region of Portugal and in the sub-region of South Beira Interior. It is bordered to the north by Penamacor, the northwest by Fundão, to the west by Castelo Branco counties, and borders Spain (Extremadura) defined to the east by the Erges river and to the south by the Tagus river [19]. It presents, in general, a Mediterranean climate (hot and dry summer and warm and rainy winter) with characteristics of continental climates (lower rainfall and higher thermal amplitudes) [20,21]. Almost 90% of the county is suitable for hunting activity [22], and this territory has one of the largest population densities of red deer in the country.

### 2.1. Sample Collection

From December 2015 to February 2016, several drive hunting actions were followed in different hunting areas and the hunted red deer heads were assessed for dipteran larvae investigation during the sanitary evaluation (referred as initial examination) performed by a veterinarian.

A feasible methodology was adopted to collect as many larvae as possible. For a transverse section on the ventral face of the neck, caudally to the branches of the mandible, the larvae collection was made at the nasopharynx level, with the main focus on pharyngeal pockets, the preferred site for L2 and L3 larvae development according to some authors [16]. The collected larvae were placed in small containers, properly identified, and stored in 70% alcohol for further identification at the laboratory. All samples collected were properly identified in relation to hunting zone, gender, and age group, which was categorised as young (<1 year), subadults (1–2 years), and adults (>2 years). Additionally, for each drive hunting action, the number of red deer hunted per hectare of each hunting area was registered (hunting bag).

### 2.2. Larvae Identification

After February 2016, with the support of the Animal Biology, Plant Biology, and Ecology laboratory of the University of Jaén, morphological identification of all larvae collected was performed according to the descriptions and keys presented by Zumpt [7] in 1965. With the aid of a magnifying glass, each larva was analysed, with the main point of differentiation being the respiratory stigmas (peritremes), which are structures that are in the most caudal part of the larva.

### 2.3. GIS Project

A Geographical Information System project (ArcGIS 10.7) was established. The GIS project was created with an Idanha-a-Nova County boundary and Parishes and hunting area limits within this county, with free information produced and provided by the Portuguese cartography authorities and by the Portuguese Institute for Nature and Forests Conservation.

The achieved results from the hunting bag and from larvae identification were used to update the GIS project.

In a second stage, geostatistical procedures were performed to map the hunting zones according to hunted animals per hectare, based on average records from 2006 to 2016 for hunting areas within Idanha-a-Nova County [23], crossed with the values of prevalence and intensity of infestation by dipteran larvae.

Because the number of infield observations was lower than 16, the Inverse Distance Weighting algorithm was used as an interpolation method [24]
$$z\;x=\overset{k}{\underset{i}{\sum }}\frac{zi}{di2}/\overset{k}{\underset{i}{\sum }}\frac{1}{di2}$$
where:*z x*—the z value to estimate at local *x**z i*—*z* value observed at local *i**di2*—squared distance from local *x* to local *i**k*—number of neighbors’ locals around local *x* to be used in the estimation*z*—variable in analysis

### 2.4. Statistical Analysis

Using IBM SPSS Statistics^®®^ 20 software (IBM Corp., Armonk, NY, USA), data were submitted to the following statistical tests:−Fisher’s exact test to analyse the significance between differences in prevalence;−Student *t*-test to assess statistical significance between differences in intensity;

A probability value (*p*-value) < 0.05 was considered statistically significant.

## 3. Results and Discussion

Of the total number of analysed larvae (*n* = 956), morphological characteristics consistent with *Pharyngomyia picta* (*n* = 795, 83.2%) and *Cephenemyia auribarbis* (*n* = 161, 16.8%) were observed. The species of the larvae identified corresponded to the larval species usually found in red deer. Although it was not verified in this study, some authors also mentioned that *Cephenemyia stimulator* [25] may occasionally occur, a species that appears normally associated with roe deer (*Capreolus capreolus*) [2].

A total of 166 red deer heads were evaluated, and 84 infested individuals were identified, as presented in Table 1.

The prevalence of infestation with one or both species, *P. picta* and *C. auribarbis*, was 50.6 ± 7.61% for a 95% confidence interval (CI). Although the larvae of these parasites were previously observed in red deer in Portugal [26], it is the first time that a value of its prevalence was estimated and species were identified. This value is in line with the expected result since this taxonomic group tends to affect a considerable number of hosts. It was close to the values obtained in the central region of Spain by De la Fuente et al. in 2000 [13] (41.89%), or in Austria by Leitner et al. in 2016 [27] (57.6%). The lowest prevalence found in the literature review was 35.19% [8], and more recently Gonzalvez et al. [3] registered 37.5% of affected animals. However, most references report a much higher prevalence than the value estimated by the present study of 85% [5], 90% [15], 94.3% [28], 98.2% [29]. 

The red deer populations of Idanha-a-Nova have a distribution mainly to the east and south of the county, near the border area with Spain, explaining why samples were only obtained from hunting areas in this geographical region (Figure 1). Figure 1 also shows that the Oestrosis prevalence was higher in the Northeast (82.4%) and the Southeast of the county (84.4%, 69.6%). This higher prevalence occurred in the hunting areas where higher population densities were observed (hunting bag).
OP = 13.844 Ln(HB) + 107.08     R^2^ = 0.347 (*p*-value = 0.0439) 
where:OP—Osteosis prevalence HB—Hunting bagLn—Neperian logarithm

This result is in accordance with the biological cycle of this parasite, since adults have a short life stage, being devoted only to finding a new host for the laying of larvae [10]. Thus, the higher the animal density in a hunting area, the more easily these diptera find the hosts before succumbing and, consequently, the higher the prevalence of infestation in the animal population of that area. The existence of shelters for the animals or the presence of greater vegetation cover in some hunting areas can influence the spread of the disease. These data have not been studied but we must take these factors into account in future work.

Additionally, proximity to Spain could also be one of the factors that justifies highest prevalence along the border [30]. This Portuguese region, with Spanish territories nearby, is known for having high densities of red deer, which can predispose them to the highest prevalence of nasopharyngeal myiasis. Coordinated actions from both countries could help to decrease this parasitosis, for example, by controlling deer populations and/or eliminating larvae found during initial examination after hunting activities. It is known that conversion into pupa may also occur while they are still in the nasal cavities of the host [1].

The mean abundance and its standard deviation (total number of larvae/total number of animals analysed) were 5.76 ± 9.36, a value similar to that obtained by Vicente et al. in 2004 [8] (5.49 ± 12.12). The mean intensity of parasitism per affected host (total number of larvae/total number of affected animals [31]) was 11.38 ± 10.47 larvae, which was similar to the recent data provided by Gonzalez et al. [3]. Nevertheless, it was very low compared with the majority of the reviewed literature studies performed during the winter months that reported 25 to 30 larvae per individual [5,15,28], or even when compared with studies performed over a full year describing approximately 15 larvae/host [8,13]. On the other hand, a study conducted between April and July 2014 in Austria [27] recorded a lower intensity (6.69 larvae/host), which could be related to the study period when L3 larvae start to leave the host to become pupae.

It is possible to speculate on the fact that only two larval stages, L2 (*n* = 219) and L3 (*n* = 737), were found in both species. By collecting the larvae that were in the nasopharynx region, the nasal sinus was not assessed, which is the place where the larvae of the first stage (L1) develop and move in the host. Another hypothesis is the small size of the L1 (maximum length: 6 mm, [7]) makes it unnoticeable during sampling under the evaluation conditions used. Moreover, the absence of L1 larvae during the collection period may have truly been the case. Adult flies are more active in the hot months (20–30 °C), becoming inactive when the environmental temperature decreases to 8–10 °C [9]. Considering the samples were collected during the winter months, the last deposition of larvae in the host might have occurred somewhere between September (20–30 °C) and mid-October (13–25 °C), according to the meteorological data of 2015 [32]. Therefore, L1 larvae may eventually have sufficient time to develop into L2. Considering these facts, it is possible to suggest this methodology as a feasible and cost-effective tool to evaluate the presence of oestridae larvae in hunted red deer to estimate the prevalence in the population, as was previously underlined by other authors [8].

It was also estimated that Oestridae larvae prevalence and its mean intensities occurred during the three months in which the study took place. The prevalence in December did not exceed 20% (17.91 ± 9.18), having increased and reached almost 90% in January (88.46 ± 8.68%), and in February decreased to about 50% (55.32 ± 14.21). Vicente et al. [8] also observed its highest value in January (71.79%). It is hard to understand the prevalence of evolution during these three months. However, this type of variation was previously reported by other authors [8,13], and could an effect from the overlapping prevalence of *P.picta* and *C. auribarbis*. Regarding the intensity of oestrid larvae per host, it started low in the first two months (7.67 ± 9.84 in December and 8.26 ± 6.62 in January) and reached its maximum in February (18.62 ± 12.78), with a value close to that reported by Vicente et al. in 2004 [8] (19.39 larvae/animal). This increasing value over time may be associated with the concomitant increase in the size of the larvae becoming more easily detectable.

The prevalence of *P. picta* (43.98 ± 11.39%) was higher than of *C. auribarbis* (19.28 ± 13.67%), which agrees with the results previously reported by Gil Collado et al. [30], Martínez and Palomares [15], Ruiz et al. [28], Bueno-de la Fuente et al. [5] and Leitner et al. [27]. The mean infestation intensity by *P. picta* was 10.89 ± 10.25 and by *C. auribarbis* it was 5.03 ± 6.83, with this difference being highly significant (t = 3.442; *p* = 0.0108). This result is in agreement with the ones obtained by several authors [14,16,26]. The achieved value is similar to that reported by Vicente et al. [8] (12.98 larvae/host), but most of the intensities reported for *P. picta* were higher than 20 larvae/host. Regarding *C. auribarbis*, some authors [15,28] reported intensity values similar to those obtained in this study.

Prevalence and mean intensity (Figure 2) of *P. picta* were higher than of *C. auribarbis* in practically the entire territory. An exception was observed in the south Rosmaninhal (Parish 17), where in one hunting area the mean intensity of *C. auribarbis* was higher. Additionally, in three HA, two in Penha Garcia (Parish 1) and one in Segura (Parish 15), infestations by *P. picta*. were exclusively found. However, in these places, fewer samples were analysed, so it was not possible to measure with reasonableness if the geographical dispersion of *P. picta* is superior to that of *C. auribarbis*. 

There were 52 pure infections of *P. picta*, 11 pure infections by *C. auribarbis,* and 21 coinfections (25% of total infections). The presence of coinfections by the two species is coincident with what was already described for nasopharyngeal myiasis in central Spain by Martínez and Palomares [15]. In terms of pure infections, the mean intensity of *P. picta* (11.50 ± 10.48) was higher than that of *C. auribarbis* (8.09 ± 10.57), with no statistically significant difference (t = 0.979, *p* = 0.3602). In concomitant infections, the same trend was found: *P. picta* (9.38 ± 9.74) and *C. auribarbis* (3.43 ± 2.94), with this difference being statistically significant (t = 2.681, *p* = 0.0315). There seems to be a dominance of *P. picta* in both types of infection. However, Vicente et al. [8] demonstrated that the intensity of *P. picta* infection was negatively affected by the presence of *C. auribarbis*. Although it was tested, no statistically significant results were achieved.

Differences in the main epidemiological variables have been described depending on the host species involved, gender, age, and climatic factors [8]. In this study, the potential differences in prevalence and mean intensity related to gender and age were also analysed.

The larval prevalence was significantly higher (X^2^ = 4.35; *p* = 0.0370) in males (62.16 ± 11.05%) than in females (41.30 ± 10.06%), which corroborates the data reported in the literature review [3,5,8]. On the other hand, the mean intensity of infestation was lower in males (9.93 ± 8.54) when compared to females (13.13 ± 12.3). However, this difference was not significant (t = 1.403, *p* = 0.1958). Several authors also found no significant differences in the mean intensity regarding gender [13,15,28]. Animals suffering immune, physiological or pathological stress may have had a greater predisposition to suffer a higher degree of infestation. Males often have a higher parasite prevalence than females, as testosterone may negatively impact immunity, and also, in the case of nasopharyngeal myiasis, there may be an anatomical explanation, as males have a larger retropharyngeal space for larval invasion [3,9]. It may also be related to the gregarious behavior of females, in opposition to the solitary life of males. Similarly to what occurs in sheep [1], when red deer can crowd and hide the muzzles through the herd, this can function as a defense mechanism against flies.

Variations in prevalence and intensity between the different age groups were also analysed. The achieved results showed that the prevalence in young (Y; *n* = 7), subadults (SA; *n* = 43), and adults (A; *n* = 116) were, respectively, 28.57 ± 33.47%, 62.79 ± 14.45%, and 47.41 ± 9.09%. It was found that the prevalence in the subadults group was higher than in the other two groups. Prevalence in the younger group was lower than that observed in subadults, with this difference being highly significant (X^2^ = 15.33; *p* = 0.00009). It was also lower, with a very significant difference (X^2^ = 6.84; *p* = 0.0089) when compared to the prevalence found in adults. However, no significant differences were found between subadults and adults (X^2^ = 2.78; *p* = 0.0954). We should remain cautious in the interpretation of these results because the sampling of young deer was smaller (*n* = 7) compared to the other age classes. Comparing the achieved results with those previously presented by some authors that used similar age division it was possible to observe some similarities. De la Fuente et al. [13] showed no significant differences between the different groups, but the younger group always displayed highest prevalence. Additionally, Ruiz et al. [28] described adults as the group with the highest prevalence. In other articles that presented a different organization for age classes, no consensus was observed, even with some supporting an increasing prevalence with age [5], and conversely, some stated that adults constitute the group with significantly lower prevalence [8].

For the mean intensity between age groups, no significant differences were observed between the age classes studied (t Y, SA = 0.889; *p* = 0.4035|t Y, A = 0.959; *p* = 0.3695|t SA, A = 0.555; *p* = 0.5962). The mean results were as follows: 4.50 larvae per affected young deer, 12.26 larvae per affected subadult, and 11.20 per affected adult. De la Fuente et al. [13], despite having observed a maximum intensity in younger deer (20.4 larvae/animal), presented values very similar to ours for the subadults (12.73) and adults (11.35). Vicente et al. [8] also described a decreasing profile of intensities depending on the increase in age. Young deer may have a lower prevalence since they stay close to females, spending most of the time hidden from predators. Additionally, immunity acquisition is expected to decrease the infestation as age increases. It is necessary to perform further sampling in more detail to better understand the effect of gender and age on nasopharyngeal myiasis. 

## 4. Conclusions

This was the first study derived in Portugal that identified the prevalence of infestation with *P. picta* and *C. auribarbis*in red deer. The values of prevalence (50.6%) and mean intensity (11.38 larvae) suggest that surveillance at the national level for this parasite is an important part of deer health management. Moreover, it could be beneficial to start raising awareness among game managers about this problem. For example, to encourage them to dispose properly of the larvae during hunting, and thereby contribute to the interruption of the life cycle of the parasites. Furthermore, it would be important to develop further studies to estimate the sublethal effect of this parasitosis. 

Since the winter months (coinciding with the hunting season) are related to the highest prevalence and infestation intensity values, it seems that the initial examination of hunted animals could be a reliable and practical tool to detect affected individuals. Using this tool, preventive measures could be implemented and their success could be followed throughout subsequent hunting seasons. It is necessary to collect and analyse further sampling in more detail to better understand the effect of gender and age. 

## Figures and Tables

**Figure 1 animals-12-01896-f001:**
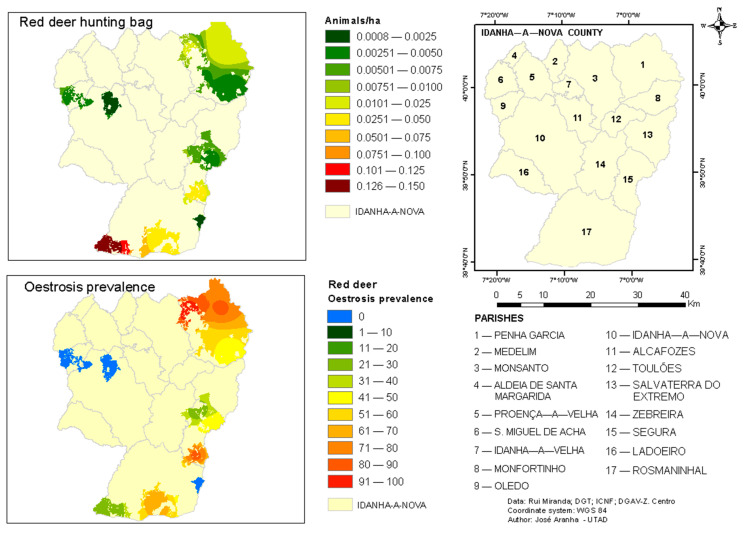
Geographical distribution of Oestrosis prevalence by hunting areas (GIS).

**Figure 2 animals-12-01896-f002:**
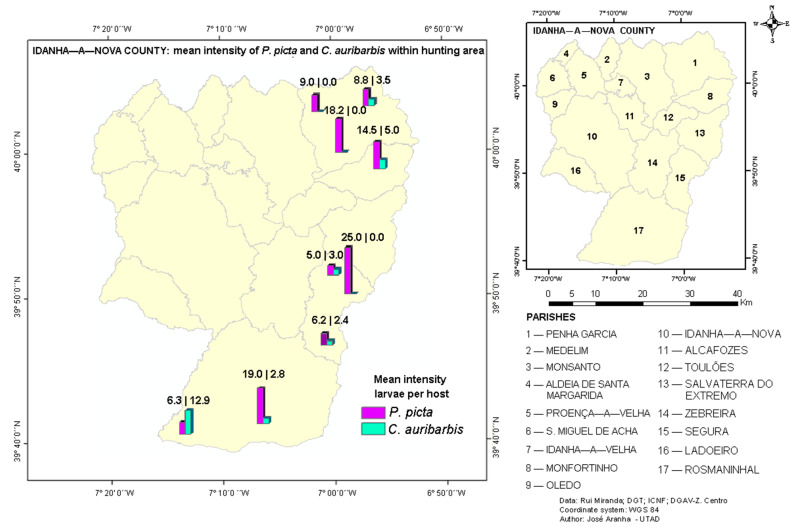
Mean intensity of *P. picta* and *C. auribarbis* in each HA.

**Table 1 animals-12-01896-t001:** Oestrosis prevalence in red deer in each hunting area (HA).

HA Code	Parishes	Analysed	Infected	Prevalence (%, CI 95%)
1	1-Penha Garcia	2	2	100.00
2	1-Penha Garcia	17	14	82.35 ± 5.80
3	1-Penha Garcia	8	4	50.00 ± 7.61
4	8-Monfortinho	9	4	44.44 ± 7.56
5	15-Segura	5	1	20.00 ± 6.09
6	15-Segura	2	1	50.00 ± 7.61
7	15-Segura	32	27	84.38 ± 5.52
8	17-Rosmaninhal	68	15	22.06 ± 6.31
9	17-Rosmaninhal	23	16	69.57 ± 7.00
Total		166	84	50.60 ± 7.61

## Data Availability

According to General Regulation on Data Protection (GRDP), regulated by Law 59/2019, all data treated within the scope of this paper are confidential.

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
