# Peer review of "First Study of Pharingomyia picta and Cephanemyia auribarbis in Wild Populations of Red Deer (Cervus elaphus) in Portugal"

_animals, 2022, doi:10.3390/ani12151896_

Round 1

Reviewer 1 Report

In the present study, the authors provide the first epidemiological data of nasopharyngeal bot flies larvae (C. auribarbis and P. picta) in red deer (Cervus elaphus) from Portugal in the current literature. They used 166 heads collected during (only) one hunting season (2015, December to 2016, February) coming from “several” drive hunting actions in “different” hunting areas at the county of Idanha-a-Nova, Castelo Branco district (Central-East Portugal).

In my opinion, this paper lacks originality with the only new insight being the (overemphasized) first scientific Portuguese report of these larvae in red deer populations. All findings are in line with previous reports from nearby counties from Spain, al also from other environments across Iberia, in spite of the existing local variations due to environmental factors.

I recognize that any research done in wildlife is significant nowadays, especially since wildlife has become of great interest to scientists at the wildlife-livestock-human-environmental interface. However, I do not feel comfortable recommending publishing this paper in this journal and in its current form. A rapid/short communication, being clearer and more concise, would be more appropriate for further versions of the manuscript.

Author Response

The changes suggested by the reviewers were made, and new images and new comments were added to the results.

Dear Reviewer

As recommended by MDPI editor we will answer to your comments and suggestions using a red letter

Comments and Suggestions for Authors

In the present study, the authors provide the first epidemiological data of nasopharyngeal bot flies larvae (C. auribarbis and P. picta) in red deer (Cervus elaphus) from Portugal in the current literature. They used 166 heads collected during (only) one hunting season (2015, December to 2016, February) coming from “several” drive hunting actions in “different” hunting areas at the county of Idanha-a-Nova, Castelo Branco district (Central-East Portugal).

In my opinion, this paper lacks originality with the only new insight being the (overemphasized) first scientific Portuguese report of these larvae in red deer populations. All findings are in line with previous reports from nearby counties from Spain, al also from other environments across Iberia, in spite of the existing local variations due to environmental factors.

I recognize that any research done in wildlife is significant nowadays, especially since wildlife has become of great interest to scientists at the wildlife-livestock-human-environmental interface. However, I do not feel comfortable recommending publishing this paper in this journal and in its current form. A rapid/short communication, being clearer and more concise, would be more appropriate for further versions of the manuscript.

Thank you for your thoughtful comments and suggestions.

We improved the GIS project and the manuscript adding additional results about hunting areas location and about red deer hunting bag spatial distribution.

We performed geostatistical analysis in order to create continuous maps about: red deer hunting bag, Oestrose prevalence, P. picta prevalence and C. auribarbis prevalence.

When analysing the new Figures, it can be realised about the spatial relationship between red deer hunting bag and prevalence as well between the two species of Oestrose.

The text was submitted to English revision and correction

We improved the Material and Methods section to

GIS project

It was created a Geographical Information System project (ArcGIS 10.7) to derive this study. The GIS project was created with Idanha-a-Nova County boundary and Parishes and hunting areas limits within this county, free information produced and provided by the Portuguese cartography authorities and by the Portuguese Institute for Nature and Forests Conservation.

The achieved results from hunting bag and from larvae identification were used to update the GIS project.

In a second stage, geostatistical procedures were used to mapping the hunting zones according to hunting bag, crossing with the values of prevalence and intensity of infestation by dipteran larvae.

Because the number of infield observations was lower than 16, it was used the In-verse Distance Weighting algorithm as interpolation method.

We improved the Results presentation section with new results.

We performed some geostatistical analysis in order to create spatial continuous maps assigned to hunting areas spatial location.

We changed 2 figures (figure 2 and 3) and introduced 1 new figure.

Now, we think that this new approach answers to your comments and recommendations.

Submission Date

01 May 2022

Date of this review

14 May 2022 13:56:01

Reviewer 2 Report

This is an interesting study on myiasis caused by oestrid larvae in the red deer in Portugal.

However, the article should improve various aspects concerning its presentation and its approach.

The English language should be revised.

The composition, in general, of the manuscript is very poor, and some punctuation marks are in the wrong place; thus, these aspects should also be revised.

All scientific names should be in italics.

References should be separated by “,” not by “;”.

INTRODUCTION

Line 90: spp. should not be in italics.

Lines 127-130 the last part of the Introduction section would be more understandable if the authors specified, as they did in the previous sentences, to which country or region the prevalence reported in these scientific references belongs.

MATERIAL AND METHODS

2.1. Sample collection

The first sentence is not needed and it is also repetitive.

2.1. SIG

In English, SIG (Sistema de Informações Geográficas) is GIS (Geographic Information System). Moreover, ArcGis 9.x is the software in which a GIS project can be created, but it is not a GIS.

What is the objective of the creation of this GIS project? Further explanations about the layers included in the project, their spatial and temporal resolution should be included.

RESULTS

If the nasal sinus was not assessed, which is the place where the L1 develops and moves in the host, the prevalence could be underestimated. Please, include this question as a limitation of the study, although the explanation given by the authors about the absence of L1 larvae is convincing.

Figure 2 is not a GIS, it is a county map of the study area with overlapped prevalence of the 9 sites prospected only.

Where is the GIS project?

Line 227: Mean abundance is the total number of larvae found / the total number of hosts analysed (not animals observed)

Line 229: The authors should define Mean Intensity.

Lines 243-257: this paragraph is confusing, it should be re-written.

Line 257-258: I do not understand the reason of this sentence, mainly because the authors did not present, in the Material and Methods section, the sample, i.e. the number of animals analysed by month, by sex, and by age.

Figure 3: Average intensity is Mean intensity

Lines 263-266: this paragraph is confusing, it should be re-written.

X2 should be written with superscript.

The authors report the values of this test in subsection 3.4. (sex and age) but not before, when previous reported prevalence were compared, for example between the prevalence of months or the two species analysed.

Moreover, the mean intensity cannot be compared using the Chi square test. To analyse mean intensity and mean abundance Student t or Anova (as parametric tests) or Mann-Whitney or Kruskal-Wallys (as non-parametric test) should be used.

In general, this section needs to be deepened a little more in the explanation, it should not be limited to a comparison with previous studies only.

CONCLUSIONS

The authors mention hat preventive measures could be implemented. This is the first time that the authors refer to this question.

If there are clear preventive measures that could be applied, the authors should explain in the discussion section which these are and how and when these could be implemented.

Author Response

The changes suggested by the reviewers were made, and new images and new comments were added to the results.

Dear Reviewer

As recommended by MDPI editor we will answer to your comments and suggestions using a red letter.

Comments and Suggestions for Authors

This is an interesting study on myiasis caused by oestrid larvae in the red deer in Portugal.

However, the article should improve various aspects concerning its presentation and its approach.

The English language should be revised.

The composition, in general, of the manuscript is very poor, and some punctuation marks are in the wrong place; thus, these aspects should also be revised.

Thank you for your thoughtful comments and suggestions.

We improved the GIS project and the manuscript adding additional results about hunting areas location and about red deer hunting bag spatial distribution.

We performed geostatistical analysis in order to create continuous about: red deer hunting bag, Oestrose prevalence, P. picta prevalence and C. auribarbis prevalence.

When analysing the new Figures, it can be realised about the spatial relationship between red deer hunting bag and prevalence as well between the two species of oestrid.

The text was submitted to English revision and correction.

All scientific names should be in italics.

References should be separated by “,” not by “;”.

Thank you for your notes

We revised the scientific names and apply for italics and we changed “;” by “,” in references presentation along the text.

INTRODUCTION

Line 90: spp. should not be in italics.

Done in the revised manuscript.

Lines 127-130 the last part of the Introduction section would be more understandable if the authors specified, as they did in the previous sentences, to which country or region the prevalence reported in these scientific references belongs.

Done in the revised manuscript, now in lines 126-130 (see the document with active track changes).

MATERIAL AND METHODS

2.1. Sample collection

The first sentence is not needed and it is also repetitive.

We agree with your comment. Sampling collections starts now by: “For this study, we adopted a methodology that was easy to apply in practical terms in the field, and that would allow us to collect as many larvae as possible.”

2.3. SIG

In English, SIG (Sistema de Informações Geográficas) is GIS (Geographic Information System). Moreover, ArcGis 9.x is the software in which a GIS project can be created, but it is not a GIS.

Yes, this was a real lack of attention.

We changed to:

“GIS project

It was created a Geographical Information System project (ArcGIS 10.7) to derive this study. The GIS project was created with Idanha-a-Nova County boundary and Parishes and hunting areas limits within this county, free information produced and provided by the Portuguese cartography authorities and by Portuguese Institute for Nature and Forests Conservation.

The achieved results from hunting bag and from larvae identification were used to update the GIS project.

In a second stage, geostatistical procedures were used to mapping the hunting zones according to hunting bag, crossing with the values of prevalence and intensity of infestation by dipteran larvae.

Because the number of infield observations were lower than 16, it was used the In-verse Distance Weighting algorithm as interpolation method.”

What is the objective of the creation of this GIS project? Further explanations about the layers included in the project, their spatial and temporal resolution should be included.

Thank you for your notes.

As previous presented, we performed some geostatistical analysis in order to create spatial continuous maps assigned to hunting areas spatial location.

We changed 2 figures (figure 2 and 3) and introduced 1 new figure.

Now, we think that this new approach answers to your comments and recommendations.

RESULTS

If the nasal sinus was not assessed, which is the place where the L1 develops and moves in the host, the prevalence could be underestimated. Please, include this question as a limitation of the study, although the explanation given by the authors about the absence of L1 larvae is convincing.

Make sense. It is now included in the conclusions of the new manuscript (lines 400-402; see the document with active track changes).

Figure 2 is not a GIS, it is a county map of the study area with overlapped prevalence of the 9 sites prospected only.

Where is the GIS project?

We agree with your comments.

As previous presented, we performed some geostatistical analysis in order to create spatial continuous maps assigned to hunting areas spatial location.

Now, we think that this new approach answers to your comments.

Line 227: Mean abundance is the total number of larvae found / the total number of hosts analysed (not animals observed)

We agree with your comment.

We check the result and confirm the presented values. We changed (total number of larvae / total number of animals observed analysed) (line 253; see the document with active track changes).

Line 229: The authors should define Mean Intensity.

The following definition was added to the main text (lines 254-255; see the document with active track changes): total number of larvae / total number of affected animals.

Furthermore, the authors add to the text the bibliographic reference from where the definition was taken (Margolis et al., 1982).

Lines 243-257: this paragraph is confusing, it should be re-written.

The paragraph was re-written. Now it is between the lines 267-281 (see the document with active track changes).

Line 257-258: I do not understand the reason of this sentence, mainly because the authors did not present, in the Material and Methods section, the sample, i.e. the number of animals analysed by month, by sex, and by age.

You are absolutely right. The sentence is out of context, so it has been deleted.

Figure 3: Average intensity is Mean intensity

Corrected in the reviewed manuscript.

Lines 263-266: this paragraph is confusing, it should be re-written.

We decided to eliminate this information since it is in the conclusion section.

X2 should be written with superscript.

Thank you for the note. We changed along the text.

The authors report the values of this test in subsection 3.4. (sex and age) but not before, when previous reported prevalence were compared, for example between the prevalence of months or the two species analysed.

The authors do not understand very well what the reviewer wants, and that is why we ask for more information so that we can correct it according to what is requested.

Moreover, the mean intensity cannot be compared using the Chi square test. To analyse mean intensity and mean abundance Student t or Anova (as parametric tests) or Mann-Whitney or Kruskal-Wallys (as non-parametric test) should be used.

In general, this section needs to be deepened a little more in the explanation, it should not be limited to a comparison with previous studies only.

We agree with your comments and appreciate your suggestion.

According to data structure in analysis, we performed:

- Fisher’s exact test to analyses significance between differences in prevalence;

- Student t-test to assess statistical significance between differences in intensity;

A probability value (p-value) < 0.05 was considered statistically significant.

We update this section with additional results about differences significance.

CONCLUSIONS

The authors mention that preventive measures could be implemented. This is the first time that the authors refer to this question.

If there are clear preventive measures that could be applied, the authors should explain in the discussion section which these are and how and when these could be implemented.

This is now included in the discussion between lines 246-251 (see the document with active track changes).

Submission Date

01 May 2022

Date of this review

14 May 2022 12:24:45

Round 2

Reviewer 1 Report

General comment:

I have suggested changing the wording to a short communication format and, in contrast, the authors have expanded the methods and results sections. I am sorry, but these larvae infestations are widely described in deer species in the scientific literature and therefore the text should be shortened, going directly to the main findings. Conclusions should be clearer and avoid speculation.

Specific comments:

The style and expressions in English are still not up to the standards of scientific writing. The writing is often confusing and difficult for readers to follow.

The introduction is very long and includes a lot of information that should be summarised or even removed. The last paragraph of this section belongs to the discussion.

Estimates of abundance, prevalence and intensity (which are the only objectives of this study) are not clearly defined and explained in the methods section.

Lack of scientific rigour in presenting results (i.e., bar graphs mix different scales/units, scientific names are not in italic in some of the figures, decimal units are very heterogeneous throughout the manuscript and colours are uninformative in figure 6, for example). 

The hunting bags added in the revised version could be useful data, but they cannot be presented in absolute terms. They need to be presented in relation to the area considered (in km2 or hectares) in order to make proper comparisons. 

Other important factors could add further value to this document, such as management practices in each hunting state, fencing, supplementary feeding, the presence of livestock and potentially the use of deworming products....

The conclusions again lack scientific rigour and do not clearly answer the proposed objectives. They are very extensive and include topics that should be moved to the introduction or discussion sections.

Author Response

Dear Reviewer

As recommended by the MDPI editor we will answer your comments and suggestions using a red letter.

Comments and Suggestions for Authors

General comment:

I have suggested changing the wording to a short communication format and, in contrast, the authors have expanded the methods and results sections. I am sorry, but these larvae infestations are widely described in deer species in the scientific literature and therefore the text should be shortened, going directly to the main findings. Conclusions should be clearer and avoid speculation.

We thank you in advance for taking the time to appreciate our work. In fact, we didn't shorten our article before because the two reviewers' views were a little different. For this reason, we asked the editors' opinion and it was concluded that it is indeed to be shortened. We shortened the document, mainly in de discussion where we focused more on the distribution of the disease. We hope you find this new document more suitable for publication.

 Specific comments:

The style and expressions in English are still not up to the standards of scientific writing. The writing is often confusing and difficult for readers to follow.

English language editing will be arranged by MDPI.

The introduction is very long and includes a lot of information that should be summarised or even removed. The last paragraph of this section belongs to the discussion.

We shortened the introduction a bit and removed the last paragraph.

Estimates of abundance, prevalence and intensity (which are the only objectives of this study) are not clearly defined and explained in the methods section.

The definitions of that concepts are presented in the results with bibliographic support. If you prefer, we can add to the methods.

Lack of scientific rigour in presenting results (i.e., bar graphs mix different scales/units, scientific names are not in italic in some of the figures, decimal units are very heterogeneous throughout the manuscript and colours are uninformative in figure 6, for example). 

We deleted some images and improved the remaining ones.

The hunting bags added in the revised version could be useful data, but they cannot be presented in absolute terms. They need to be presented in relation to the area considered (in km2 or hectares) in order to make proper comparisons.

You are completely right. Now the hunting bags are presented in relation to the area (hunting bag/hectares).

Other important factors could add further value to this document, such as management practices in each hunting state, fencing, supplementary feeding, the presence of livestock and potentially the use of deworming products....

Fencing is not common in this area. Supplementary feeding occurs. We added the following information to the text: “The existence of shelters for the animals or the presence of greater vegetation cover in some hunting areas can influence the spread of the disease. These data have not been studied but we must take these factors into account in future work.”

The conclusions again lack scientific rigour and do not clearly answer the proposed objectives. They are very extensive and include topics that should be moved to the introduction or discussion sections.

We rearranged the conclusions.

Reviewer 2 Report

The Ms has been improved by the authors and almost all my comments and suggestions have been addressed. However, there are few things that need to be corrected.

sp. and spp. should not be in italics.

Better than Margolis et al. (1982), use Bush et al. (1997):

Bush, A.O.; Lafferty, K.D.; Lotz, J.M.; Shostak, A.W. Parasitology meets ecology on its own terms: Margolis et al. revisited. J. Parasitol. 1997, 83, 575–583.

In Margolis et al. (1982), mean abundance is not clearly defined.

Along the document, it is better to report the real value of P, concerning χ2 and t, than P is >0.05, or <0.05 or <0.001, if the authors have retained these values of P.

Concerning the question that the authors did not understand, it is ok, it does not matter.

Author Response

Dear Reviewer

As recommended by the MDPI editor we will answer your comments and suggestions using a red letter.

Comments and Suggestions for Authors

The Ms has been improved by the authors and almost all my comments and suggestions have been addressed. However, there are few things that need to be corrected.

Thank you for your feedback.

English language editing will be arranged by MDPI.

  1. and spp. should not be in italics.

This was already done in the previous review.

Better than Margolis et al. (1982), use Bush et al. (1997):

Bush, A.O.; Lafferty, K.D.; Lotz, J.M.; Shostak, A.W. Parasitology meets ecology on its own terms: Margolis et al. revisited. J. Parasitol. 1997, 83, 575–583.

In Margolis et al. (1982), mean abundance is not clearly defined.

Thank you for your suggestion. In the article you mention, the definitions are definitely clearer. We followed your idea and cited Bush et al. instead of Margolis et al..

Along the document, it is better to report the real value of P, concerning χ2 and t, than P is >0.05, or <0.05 or <0.001, if the authors have retained these values of P.

This suggestion was added to the new document.

Concerning the question that the authors did not understand, it is ok, it does not matter.

Thank you anyway.